# Challenges in Diagnosis of Uretero–Arterial Fistulas after Complex Pelvic Oncological Procedures—Single Center Experience and Review of the Literature

**DOI:** 10.3390/diagnostics12081832

**Published:** 2022-07-29

**Authors:** Cristian Surcel, Cristian Mirvald, Robert Stoica, Vasile Cerempei, Isabel Heidegger, Apostolos Labanaris, Igor Tsaur, Catalin Baston, Ioanel Sinescu

**Affiliations:** 1Centre of Uronephrology and Renal Transplantation, Fundeni Clinical Institute, 011455 Bucharest, Romania; drsurcel@gmail.com (C.S.); stoicarobert@gmail.com (R.S.); vcerempei@yahoo.com (V.C.); drcbaston@gmail.com (C.B.); urologiefundeni@gmail.com (I.S.); 2Faculty of General Medicine, “Carol Davila” University of Medicine and Pharmacy, 020021 Bucharest, Romania; 3Department of Urology, Medical University Innsbruck, 6020 Innsbruck, Austria; isabel-maria.heidegger@i-med.ac.at; 4Interbalkan Medical Center, 555 35 Thessaloniki, Greece; labanaris@web.de; 5Department of Urology and Pediatric Urology, University Medicine Mainz, 55131 Mainz, Germany; igor.tsaur@unimedizin-mainz.de

**Keywords:** uretero–arterial fistula, diagnostic, arteriography, computed tomography, urinary diversion

## Abstract

**Introduction:** Uretero–arterial fistula (UAF) represents a rare condition that manifests as massive or intermittent hematuria and requires collaboration between a urologist, vascular surgeon and interventional radiologist. In this article, we present our experience with UAF diagnosis, treatment pathways and the results of a nonsystematic review of the literature published in the last decade regarding modern diagnostic procedures. **Material and method:** We analyzed the clinical data of nine consecutive patients from our institution diagnosed with UAF in the interval of 2012–2022 who underwent open or endovascular surgical treatment. We reviewed patient characteristics, diagnoses and treatment pathways. The literature search resulted in 14 case series, published from 2012 to 2022, describing a total of 670 cases of UAF. **Results:** The mean age of patients in our cohort was 65.3 years (IQR: 51–79). UAFs were more common in women (77.7%). All patients presented a history of surgical intervention and ir-radiation for pelvic malignancy with permanent ureteric stenting. Overall, 88.8% of patients had urinary diversion, either via ileal conduit or cutaneous ureterostomy. The most common clinical manifestation of UAF was gross hematuria with or without clots accompanied by flank pain due to stent obstruction, while three patients presented with hypovolemic shock. Angiography represents the best option for diagnosis, followed by angioCT, with a sensitivity of 59.83% and 47.01%, respectively. There is no definitive imaging modality associated with high accuracy in detecting UAF and negative findings do not exclude the disease. In emergency cases with massive bleeding, surgical exploration remains the most appropriate management option for both diagnosis and treatment. Endovascular stent graft placement is preferred over open surgery in stable hemodynamic patients. **Conclusions:** Uretero–arterial fistulas represent a life-threatening complication and must be treated with great awareness. Angiography represents the best modality for diagnosis, followed by computed tomography. However, there is no definitive imaging modality and, in some cases, open approach remains the only option for diagnosis and treatment.

## 1. Introduction

Uretero–arterial fistula (UAF) represents a rare condition that manifests as massive or intermittent hematuria and requires collaboration between a urologist, vascular surgeon and interventional radiologist [1,2,3]. Specific-cause mortality varies between 7–90%, depending on type of treatment (open vs endovascular), hemodynamic status of the patient and hospital infrastructure [1,4,5,6]. UAF incidence is increasing due to improved survival after aggressive management of pelvic tumors that involves large surgical resections, urinary diversions and radiotherapy [5,7,8]. Diagnostic pathways and treatments are very heterogeneous, since most of the published articles are limited to case series or reports [1,9,10,11].

In this article, we present our experience with the management UAF treated by open or endovascular treatment based on the specific risk profile of the patients.

## 2. Materials and Methods

We analyzed the clinical data of 9 consecutive patients from our institution diagnosed with UAF in the interval of 2012–2022 who underwent open or endovascular surgical treatment. We reviewed patient characteristics, diagnosis and treatment pathways. Subsequently, a nonsystematic review of the literature published in the last decade regarding modern diagnostic procedures was also performed. Two different syntaxes of search terms were used: ureteroarterial, aortoarterial, ureteral-arterial, ilioureteric, artery-ureteral, ureteral-ileal, arterio-ureteral, ureteroiliac, fistula, diagnostic and imaging. Only reports of case series and reviews which included more than 5 patients were selected. We excluded from analysis articles describing fistulas between the ureter and iliac vein, bladder fistulas, primary aneurysm of any origin (aortic, renal, hypogastric, iliac) or papers with no data regarding diagnostic imaging. The literature search resulted in 14 patients-series, published from 2012 to 2022, describing 670 cases of UAF.

## 3. Results

The mean age of patients in our cohort was 65.3 years (IQR: 51–79) and UAFs were more common in women (77.7%). All patients presented a history of surgical intervention and ir-radiation for pelvic malignancy with permanent ureteric stenting. Overall, 88.8% of patients had urinary diversion (either ileal conduit or cutaneous ureterostomy) and one patient had chronic indwelling ureteral stents for ureteric stenosis after radiotherapy for cervical cancer. The most common clinical manifestation of UAF was gross hematuria with or without clots accompanied by flank pain due to stent obstruction, while three (33.33) patients presented with hypovolemic shock. Mean hemoglobin at diagnosis was 7.24 mg/dL, (interval 5.1–10.8) and mean serum creatinine was 1.4 mg/dL (interval 0.6–2.4) (Table 1).

In all patients, the UAF was initially assessed by contrast-enhanced computed tomography (CT) that identified the fistula only in five cases (55.5%). Angiography was performed in hemodynamic stable patients +/− ureteric stent removal during the procedure and was positive in five out of seven patients (Figure 1). In one case, the suspicion of UAF was raised during cystoscopic change of ureteric stent when a pulsating bleeding was observed after stent removal. Retrograde pyelography was positive, confirming a fistula between the left ureter and common iliac artery (CIA).

The most common UAF location was between the left ureter and left common iliac artery, which occurred in 66.6% of cases. One patient presented a fistula between abdominal aorta and left ureter, and one case developed metacronous UAF between right ureter–right CIA at 9 months after endovascular stenting for a fistula between left ureter and left CIA (Table 2).

The management strategy (open surgery or endovascular stenting) was determined individually based on the specific risk profile of each patient, hemodynamic status, location of the fistula and availability of endovascular treatment.

Initial treatment of arterial defect consisted in endovascular stenting in 66.66% of cases (Figure 2). An open approach of UAF was performed in unstable patients and in cases where radiologic access was unavailable or ineffective. Arterial breach was patched in two cases while CIA ligation and femoral-to-femoral artery extra-anatomic vascular reconstruction was performed in 22.2% of patients. In 44.44% of cases, additional procedures were needed after initial treatment. Two cases required restenting due to persistence of hematuria, and CIA ligation with femoral-to-femoral artery extra-anatomic vascular reconstruction were performed due to recurrent bleeding and hemorrhagic shock in one patient.

Overall, 77.77% of patients underwent percutaneous nephrostomy with or without ureteric ligation during open surgery. In two cases, nephrectomy was needed as initial treatment for ureteric fistula, while deferred nephrectomy was indicated in one case due to pyonephrosis with MDR bacteria (Table 3).

Follow-up ranged from 4 months to 6.5 years. Overall mortality in our cohort was 44.4%, with one death as a direct result of UAF due to significant aortic bleeding. One patient developed a metachronous fistula on the contralateral side between right ureter- right CIA that was initially treated with a stent graft, and afterwards, CIA ligation with femoral-to-femoral artery extra-anatomic vascular reconstruction were performed due to recurrent bleeding (Table 3).

## 4. Discussions

Uretero–arterial fistulas represent a late complication after pelvic surgery and radiotherapy due to chronic inflammation that creates fibrous uretero–vascular adhesions [5,12]. The mechanical friction of the pulsatile artery in direct contact with the ureter combined with chronic indwelling stenting represents the main mechanism described in the formation of UAFs [1,2,3].

UAF usually involves left common iliac artery, but can also incorporate aorta, external or internal iliac vessels [4,8,10,11]. Although the reported mortality rate has recently decreased mainly to endovascular treatments, UAF remains a potentially life-threatening condition, especially in cases with large fistulas and delayed treatment [6,7,8].

The incidence of this entity is increasing due to improved survival after complex pelvic oncological surgery, higher radiation doses and frequent usage of ureteric stents for urinary diversion or extrinsic stenosis. It is difficult to estimate the real occurrence of this disease considering that an important number of UAFs will remain unrecognized due to the negative findings of imaging procedures [5].

The definitive diagnosis of UAF is often challenging and requires a high index of clinical suspicion in a patient presenting with hematuria and a history of previous extensive pelvic surgery, external beam radiotherapy and chronic ureteric stenting. In addition, the optimal management for this condition is complex and necessitates a close collaboration between a urologist, vascular surgeon and interventional radiologist. The presence of clots in the renal cavities may suggest renal bleeding instead of UAF, leading to un-necessary diagnostic tests such as renal arteriography or nephrectomy of the involved unit. Moon et al. reported a case with bilateral renal artery embolization prior to UAF diagnostic and treatment [13].

Contrast-enhanced computed tomography with multiplanar reconstructions provides an excellent morphologic evaluation and, although it revealed the presence of UAF in ~50% of patients in our cohort, it raises the suspicion of fistula in cases where there is a very close contact between the ureter and a large artery. In addition, a CT angiography is useful in ruling out other causes of hematuria and to evaluate the perfusion status of the pelvis or lower limbs [9,14,15]. Removal of the ureteric stent can increase the rate of positive diagnosis, but this maneuver cannot be performed in radiological departments in patients with an ileal conduit or a double J stent and may induce life-threatening bleeding during stent manipulation [11,15,16].

Although arteriography presents the highest sensitivity in detecting a UAF, it should not be performed only for diagnostic purposes due to its invasive nature and because it may worsen the hematuria [1,11,17]. A direct image can be seen in cases with large or complex fistulas, but most often, arteriography can detect indirect signs of UAFs such as a nipple or pseudoaneurysm of the arterial wall at the crossing of the ureter, which is suggestive for the entry point of the fistula [2,18]. Angiography can be used as a treatment modality in cases with positive diagnostic but also in patients with negative findings but with indirect signs of UAF. In our cohort, a stent graft was placed on the right CIA in a case with recurrent hematuria after a previous endovascular treatment for fistula between the left ureter and left CIA, even though the arteriography did not reveal an extravasation of contrast media.

Provocative angiography with removal of the ureteric stent during injection of the contrast media or catheter-directed mechanical friction of the arterial wall can increase diagnostic sensitivity by up to 92.9% [19,20]. However, this maneuver is highly dangerous and should be performed only in selected cases. Positioning of a balloon catheter, either in the ureter or artery, may prevent a massive bleeding after the procedure [11].

Retrograde pyelography and cystoscopy can be used for diagnosis in patients with permanent JJ stenting [1,3,4,5]. The administration of thrombolytics or high-pressure ureteropielography as provocative measures can increase the rates of positive diagnosis [21]. This procedure revealed contrast leakage at the crossing with the left common iliac artery in one case with a JJ stent, but maneuver was rarely used in our cohort since the majority of our patients presented urinary diversion, such as cutaneous ureterostomy or iliac conduit.

There is no definitive imaging modality or negative findings in order to exclude the diagnosis of UAF. In some cases, surgical exploration remains the only diagnostic procedure available, even in the modern era (Table 4).

Historically, open surgery was the main treatment for UAF [3,9,11,28]. This approach presents increased perioperative morbidity and mortality due to intense fibrosis caused by previous surgery and ir-radiation, significant bleeding, difficulties in detecting and repairing the arterial defect and the need for auxiliary procedures such as femoral-to-femoral artery extra-anatomic vascular reconstruction [2,28]. However, in cases with life threatening hematuria, complex fistulas with pelvic abscesses and in hospitals with no interventional radiological department, open surgery remains a valid option in the management of UAF. In our cohort, only one UAF related death was recorded after open surgery in a patient with a fistula between the left ureter and abdominal aorta.

The endovascular approach has gained popularity due to its efficacy and safety profile [1,5,7,8]. In addition to the early recovery, the endovascular occlusion of the fistula while maintaining the arterial flow decreases the usage of revascularization procedures, which are often required after open surgery [27]. In addition, it eliminates the need for ureteric surgery, which, in some cases, may lead to the loss of the involved renal unit [22,27,29]. Although endovascular stent-graft coverage of the arterial defect is currently the preferred treatment option, this approach does not cure the fistula, which may result in graft infection or recurrent hematuria due to subintimal backflow [5,25,30]. In our cohort, 22.22% of cases required restenting due to recurrent bleeding and, in one case, right CIA ligation with femoral-to-femoral artery extra-anatomic vascular reconstruction were performed due to significant hematuria and hemorrhagic shock (Figure 3).

Our study presents several limitations that are unavoidable due to the rarity of this disease, small number of patients included and retrospective nature of the studies included in our review. Due to these inherent factors, the results of our study cannot be generalized. Multicentric prospective studies are needed in order to provide strong, evidenced-based recommendations and treatment guidelines.

In addition, the management pathway was not uniform in our cohort due to the variations in risk factors, type of urinary diversion and hemodynamic status of the patients, leading to a biased outcome, especially in the open surgery group.

## 5. Conclusions

Uretero–arterial fistulas represent a life-threatening complication and must be treated with great awareness. Angiography represents the best modality for diagnosis, followed by computed tomography. However, there is no definitive imaging modality and, in some cases, open approach remains the only option for diagnosis and treatment.

## Figures and Tables

**Figure 1 diagnostics-12-01832-f001:**
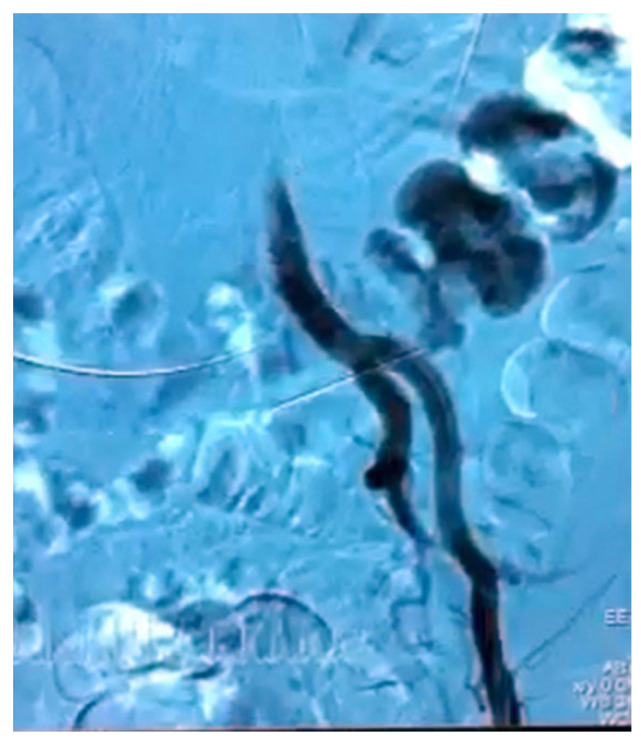
Arteriographic aspect of complex fistula between left ureter, left external iliac artery and descending colon.

**Figure 2 diagnostics-12-01832-f002:**
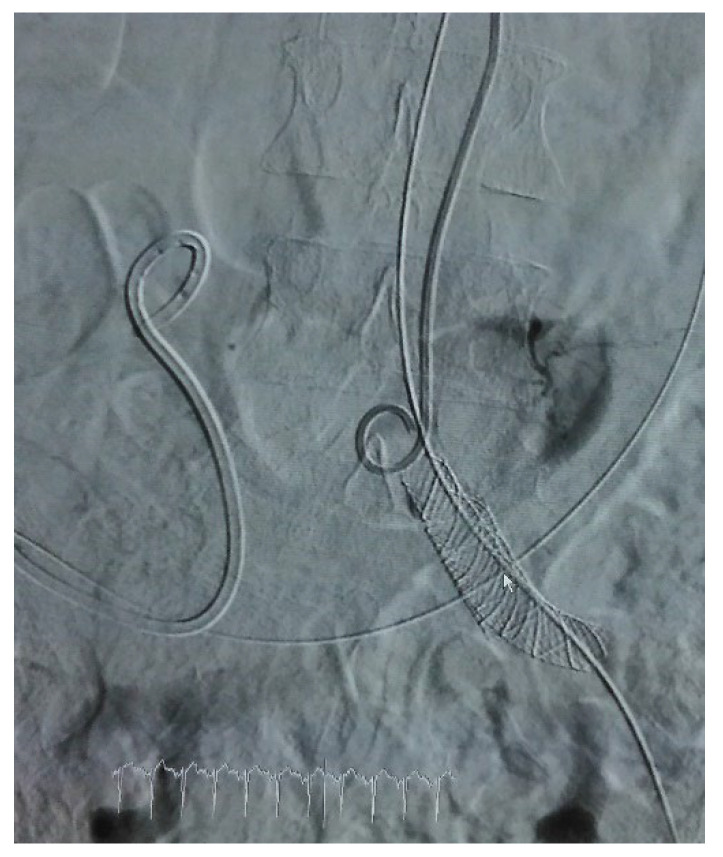
Arteriographic aspect after stent graft stenting of left CIA.

**Figure 3 diagnostics-12-01832-f003:**
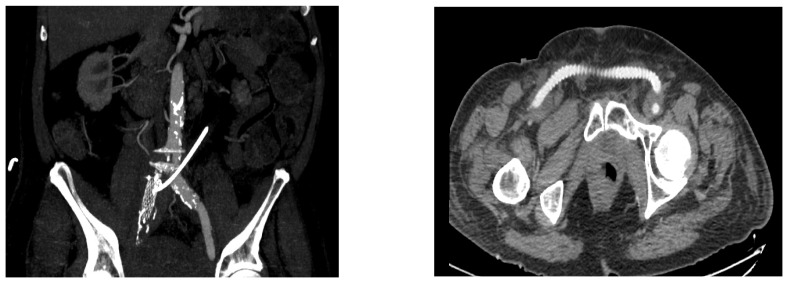
(**Right**) common iliac artery ligation with femoral-to-femoral artery extra-anatomic vascular reconstruction after failure of endovascular treatment—Angio CT aspect ((**left**): maximum intensity projection reconstruction describing the absence of contrast media in the right common iliac artery after ligation following failure of endovascular approach; right: axial section of a femoral-to-femoral artery extra-anatomic vascular reconstruction).

**Table 1 diagnostics-12-01832-t001:** Demographic data of patients from our cohort (UTI—urinary tract infection, EBRT—external beam radiotherapy, UAF—uretero–arterial fistula).

Mean Age (Yrs, IQR)	65.3 (51–79)
Sex	No, %
Male	2 (22.22)
Female	7 (77.77)
Presence of infection	
No	1 (11.11)
Multidrug resistant UTI	6 (66.66)
Permanent ureteric stenting	9 (100)
Mean time of permanent stenting to UAF (months, inverval)	26.8 (8–42.1)
Type of surgery for pelvic cancer	
Cervical cancer	
Hysterectomy	1 (11.11)
Anterior exenteration with cutaneus ureterostomy	6 (66.66)
Bladder cancer	
Cystectomy with ileal conduit	1 (11.11)
Colorectal cancer	
Total exenteration with ileal conduit	1 (11.11)
Urinay diversion	
Cutaneus urinary diversion	6 (66.66)
Ileal conduit	2 (22.22)
Radiotherapy	
EBRT only	2 (22.22)
EBRT + brachytherapy	7 (77.77)
Median time from radiotherapy to UAF (months)	34.6 (11–54.7)
Clinical symptoms	
Hematuria	9 (100)
Flank pain	6 (66.66)
Hypovolemic shock	3 (33.33)
Mean Hemoglobin at diagnostic (mg/dL, interval)	7.24 (5.1, 10.8)
Mean serum creatinine at diagnostic (mg/dL, interval)	1.4 (0.6, 2.4)

**Table 2 diagnostics-12-01832-t002:** Pretreatment diagnostic imaging sensitivity and location of UAF fistula (CT—computed tomography, CIA—common iliac artery, EIA—external iliac artery, UAF—uretero–arterial fistula).

Diagnostic Imaging Procedures	No, %
CT scan	5/9 (55.55)
Angiography	5/7 (71.42)
Retrograde pyelography	1/9 (11.11)
**Fistula location**	
Left ureter–left CIA	6/9 (66.66)
Left ureter–abdominal aorta	1/9 (11.11)
Left ureter–EIA–descending colon	1/9 (11.11)
Right ureter–right CIA	1/9 (11.11)
Metacronous UAF	1/9 (11.11)

**Table 3 diagnostics-12-01832-t003:** Pretreatment diagnostic imaging, treatment options and outcomes of patients in our cohort (CT—computed tomography; Angio—Angiography; CIA—common iliac artery; FFC—femorofemoral crossover bypass).

		Pretreatment Diagnostic Imaging									
	Year	CT	Angio	Other	Type of Urinary Diversion	UAF Location	Arterial Treatment	Ureteral Treatment	Additional Procedures	Follow-Up (Months)	Status
1	2012	Negative	Positive		USC	Left ureter- left CIA	Endovascular stent	Nephrostomy	endovascular restenting	7	Dead
2	2013	Negative	N/A		USC	Left ureter- Aorta	Aortic patch	Nephroureterectomy		1	Dead
3	2015	Positive	Positive		USC	Left ureter- left EIA+ colon	EIA ligation and FFC	ureter ligation +Nephrostomy	Left colectomy	78	Dead
4	2016	Positive	Positive		USC	Left ureter- left CIA	CIA ligation and FFC	ureter ligation +Nephrostomy		64	Alive
5	2018	Positive	Negative		USC	Left ureter- left CIA	Endovascular stent	Nephrostomy	Nephrectomy	32	Alive
6	2019	Negative	N/A	UPR	None	Left ureter- left CIA	Patch angioplasty	Nephroureterectomy		27	Alive
7	2020	Positive	Negative		Bricker	Left ureter- left CIA	Endovascular stent	Nephrostomy	endovascular restenting		
	2021	Negative	Negative			Right ureter- right CIA	Endovascular stent	Nephrostomy	Ureter ligation +CIA ligation and FFC	4	Dead
8	2021	Negative	Positive		Bricker	Left ureter- left CIA	Endovascular stent	Nephrostomy		10	Alive
9	2021	Positive	Positive		USC	Left ureter- left CIA	Endovascular stent	Nephrostomy		8	Alive

**Table 4 diagnostics-12-01832-t004:** Diagnostic test sensitivity of different imaging procedures used for diagnosis of UAF (pts—patients, CT—computed tomography, angio—angiography, URS—ureteroscopy).

Article, Year	No of Pts	Cystoscopy	CT	Angio	Pyelography	URS	Open Surgery
Malgor, 2012 [6]	20	11/14	3/8	12/14			
Okada, 2013 [22]	11		6/11	6/11			
Hong, 2016 [7]	6		4/6		1/6		1/6
Schneider, 2016 [23]	5		2/4	2/5	2/5		1/5
Das, 2016 [24]	61	26/34	13/36	55/76	16/24		42/45
Heers, 2018 [3]	24		5/14	9/23			
Titomihelakis,2019 [25]	5	1/5		5/5	1/5	2/5	
Massmann, 2020 [26]	5	1/5	5/5	5/5			
Simon, 2021 [27]	17		2/10	3/17			
Omran, 2021 [4]	25	17/25	7/23	4/21			
Ghouti, 2021 [10]	6		2/5	2/5			
Matsunaga, 2021 [11]	40		3/13	15/27	4/11	1/6	4/4
Kamphorst, 2022 [1]	445	40/142	68/141	169/272	60/118	40/142	
Our study, 2022	9	1/9	5/9	5/7	1/9		
Overall sensitivity	679	97/239 (40.58%)	134/285 (47.01%)	292/488 (59.83%)	85/178 (47.75%)	43/153 (28.1%)	48/60 (80%)

## Data Availability

The data presented in this study are available on request from the corresponding author.

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
