# Peer review of "Challenges in Diagnosis of Uretero–Arterial Fistulas after Complex Pelvic Oncological Procedures—Single Center Experience and Review of the Literature"

_diagnostics, 2022, doi:10.3390/diagnostics12081832_

Round 1

Reviewer 1 Report

The manuscript is interesting because of the subject it deals with, which is of great importance in healthcare and also because it requires the teamwork of different specialists due to its surgical and healthcare complexity. Naturally, there are more scientifically sound methodologies, such as a multicentre study, and the authors are aware of this, given that they analyse the different series published in the last decade from 2012 to 2022 by means of a classic non-systematic review; but the evidence and definitive results are often obtained through small transcendental studies such as the one addressed in this work.

Recommendation to the authors, include a paragraph on the limitations (weaknesses) and on the strengths of the study. 

Author Response

We thank the reviewer for the positive comments and careful review.

We modified the last paragraphs from “Discussion” section to broaden the limitations of our study in order to include the reviewers’ suggestions. The sentences from page 7 line 222 are now as following: “Our study presents several limitations that are unavoidable due to the rarity of this disease, small number of patients included and retrospective nature of the studies included in our review. Due to these inherent factors, the results of our study cannot be generalized. Multicentric prospective studies are needed in order to provide strong evidenced-based recommendations and treatment guidelines. In addition, the management pathway was not uniform in our cohort due to the variations in risk factors, type of urinary diversion and hemodynamic status of the patients, leading to a biased outcome, especially in the open surgery group”.

Reviewer 2 Report

This is a nice review of a rare but important condition. The overall diagnostic approach is presented beautifully with examples. 

Author Response

We thank the reviewer for the positive comment and careful review.

Reviewer 3 Report

Surcel et al provided important information about the current challenges in diagnosis of uretero-arterial fistulas after complex pelvic oncological procedures. I have some suggestions which should be addressed:

1. Line 38 in abstract should be rephrased in a better way.

2.  Line 150 in discussion should be rephrased.  

3. Figure 3 has two figures. In the figure legend both the figures should be clearly explained.

4. Authors in this manuscript selected only case series and reviews which included more than 5 patients. As Uretero-arterial fistula is a rare condition, is it not worth to include also reviews and cases having less than 5 cases.

Author Response

Surcel et al provided important information about the current challenges in diagnosis of uretero-arterial fistulas after complex pelvic oncological procedures. I have some suggestions which should be addressed:

  1. Line 38 in abstract should be rephrased in a better way.

Our response: We thank the reviewer for his suggestion

Line 38 in abstract has been rephrased from: “In some cases, surgical exploration remains the only diagnostic procedure available, even in the modern era” to “In emergency cases with massive bleeding, surgical exploration remains the most appropriate management option, for both diagnostic and treatment “.

  1. Line 150 in discussion should be rephrased.  

Our response:

We thank the reviewer for his suggestion. The paragraph: “The definitive diagnosis of UAF is often challenging are requires a high index of clinical suspicion in a patient with previous extensive pelvic surgery, radiotherapy and ureteric stenting” was rephrased as following: The definitive diagnosis of UAF is often challenging and requires a high index of clinical suspicion in a patient presenting with hematuria and a history of previous extensive pelvic surgery, external beam radiotherapy and chronic ureteric stenting.

  1. Figure 3 has two figures. In the figure legend, both the figures should be clearly explained.

Our response:

We inserted the legend of figure 3: (left: Maximum intensity projection reconstruction describing the absence of contrast media in the right common iliac artery after ligation following failure of endovascular approach; Right: axial section of a femoral-to-femoral artery extra-anatomic vascular reconstruction)

  1. Authors in this manuscript selected only case series and reviews which included more than 5 patients. As Uretero-arterial fistula is a rare condition, is it not worth to include also reviews and cases having less than 5 cases.

We agree with the reviewer. Since the aim of our study was to evaluate the accuracy of different imaging diagnostic procedure, the sensibility could not be calcutated if we would have included, for example, case presentations.